# Graphemic and Semantic Pathways of Number–Color Synesthesia: A Dissociation of Conceptual Synesthesia Mechanisms

**DOI:** 10.3390/brainsci12101400

**Published:** 2022-10-17

**Authors:** Shimeng Yue, Lihan Chen

**Affiliations:** 1School of Psychological and Cognitive Sciences, Peking University, Beijing 100871, China; 2Beijing Key Laboratory of Behavior and Mental Health, Peking University, Beijing 100871, China; 3National Engineering Laboratory for Big Data Analysis and Applications, Peking University, Beijing 100871, China; 4Key Laboratory of Machine Perception (Ministry of Education), Peking University, Beijing 100871, China

**Keywords:** synesthesia, conceptual synesthesia, synesthetic priming paradigm, graphemic and semantic pathways, representation formats, time course

## Abstract

Number–color synesthesia is a condition in which synesthetes perceive numbers with concurrent experience of specific, corresponding colors. It has been proposed that synesthetic association exists primarily between representations of Arabic digit graphemes and colors, and a secondary, semantic connection between numerosity and colors is built via repeated co-activation. However, this distinction between the graphemic and semantic pathways of synesthetic number–color connection has not been empirically tested. The current study aims to dissociate graphemic and semantic aspects of color activations in number–color synesthesia by comparing their time courses. We adopted a synesthetic priming paradigm with varied stimuli onset asynchronies (SOAs). A number (2–6, prime) was presented in one of three notations: digit, dice, or non-canonical dot pattern, and a color patch (target) appeared with an SOA of 0, 100, 300, 400, or 800 ms. Participants reported the color as quickly as possible. Using the congruency effect (i.e., shorter reaction time when target color matched the synesthetic color of number prime) as an index of synesthetic color activation level, we revealed that the effect from the graphemic pathway is quick and relatively persistent, while the effect from the semantic pathway unfolds at a later stage and is more transient. The dissociation between the graphemic and semantic pathways of synesthesia implies further functional distinction within “conceptual synesthesia”, which has been originally discussed as a unitary phenomenon. This distinction has been demonstrated by the differential time courses of synesthetic color activations, and suggested that a presumed, single type of synesthesia could involve multiple mechanisms.

## 1. Introduction

Synesthesia is a condition in which individuals (synesthetes) automatically perceive certain stimuli with experiences of additional properties, such as seeing achromatic words with certain colors. The physically present stimulus that elicits synesthesia (e.g., word) is termed the “inducer”, and the additional experience (e.g., color) is termed the “concurrent” [1]. In particular, in general grapheme–color synesthesia, inducers are graphemes (e.g., letters or digits), and concurrents are colors, while in number–color synesthesia, inducers are numbers and concurrents are colors.

To quantitatively measure synesthesia, common behavioral paradigms have been developed, including the synesthetic priming paradigm [2,3]. In a typical priming task, the prime and the target were presented consecutively, and the participant responded upon the target (e.g., pressing keys to report whether a number is larger or smaller than 5 [4,5]). The performance could be influenced by the task-irrelevant, preceding stimulus (prime). When certain attributes of a prime (e.g., its shape) provide congruent information with that associated with the target, the participant’s response is facilitated, indexed by a higher accuracy and a shorter reaction time (RT). On the contrary, when the prime conveys conflicting information, response is interrupted, resulting in a lower accuracy and a longer RT. This paradigm has demonstrated an automatic processing of the manipulated attribute of the prime and, hence, its interference with the processing of a given target. This interference can be quantified by calculating the difference in RTs between congruent and incongruent conditions (hereafter termed the *congruency effect*). Specifically, in priming paradigms for grapheme– and number–color synesthesia, a grapheme or number prime facilitates or interrupts a synesthete’s response to a color target. For instance, for a synesthete who associates the number 5 with color red, a “5” prime will lead to quicker and more accurate color-naming of a red color patch than a green one. In contrast, naïve non-synesthetes (people who do not experience synesthesia and who have not been trained with inducer–color parings for specific research purposes) show no congruency effect in such tasks.

Studies on number–color synesthesia have found that numbers automatically elicit color processing when presented in various notations, such as Arabic digits, dot patterns, and finger counts [2,6,7], demonstrating that color concurrents have been evoked regardless of variances in perceptual forms of inducers. Mathematical operations (e.g., “5 + 2”) also facilitated response to colors corresponding to the result (e.g., 7) [8,9], which means that an inducer (“7”) does not even need to be physically present to elicit the concurrent color. Together, these findings suggest that a high-level, conceptual representation of numbers is linked to color in the synesthetes’ brains (though this may not be the only source of synesthetic connection). Early theorizations of synesthesia distinguished between “lower” and “higher” types of synesthesia. The former was thought to depend on low-level perceptual properties of the inducer, and the latter was based on semantic or conceptual attributes [10,11]. Under this framework, number–color synesthesia underscores the “higher” aspect of synesthesia, since it exists regardless of how the number is presented [6,7,12,13]. More recently, Chiou and Rich [14] proposed a conceptual-mediation model as the unifying mechanism for synesthesia, suggesting that inducers and concurrents are linked on the conceptual (“higher”) level. Temporal aspects in synesthesia have been addressed by neurophysiological studies to tease apart the underlying “perceptual” vs. “conceptual” mechanisms. For instance, Brang et al. [15,16] found early (100–200 ms) event-related potential (ERP) differences in synesthesia and other mental process such as imagery, favoring the “perceptual” nature of synesthesia, while Teichmann et al. [17] argued for a “conceptual” nature of synesthesia, given that they had decoded colors from late (300–400 ms) magnetoencephalogram (MEG) signals.

However, most number–color synesthetes report no subjective color experience for numbers in less common or non-graphemic forms (e.g., dots) [6,7,18]. Accounting for this, Berteletti et al. [7] proposed a cognitive model of number–color synesthesia, and Gertner et al. [18] extended it to other types of synesthesia with number as inducers. They suggested that in the first place, Arabic digits, as symbolic graphemes, induce color experiences. When synesthetes observe a digit, they extract its semantic meaning (a numerosity) and meanwhile experience a color. Repeated simultaneous activations of paired numerosity and color representations establish the connections between them through associative learning mechanisms. The primary-digit–color connection is “synesthesia”, and the secondary, learned numerosity–color connection is termed “pseudo-synesthesia”. Pseudo-synesthesia activates color representations and produces synesthesia-like behavioral results, but this activation is usually too weak to evoke conscious color experiences. Note that graphemes (with emphasis to their identities rather than low-level features) are considered conceptual entities in the theorization of synesthesia [14], and semantic processing also operates on the conceptual level. Therefore, the proposed distinction between “synesthesia” and “pseudo-synesthesia” is indeed a distinction of processes under the umbrella of conceptual/higher synesthesia. To avoid confusion, we hereafter use the terms “graphemic pathway” and “semantic pathway” of synesthesia to refer to “synesthesia” and “pseudo-synesthesia” in the model by Berteletti et al [7], and use the term “synesthesia” for the phenomenon as a whole (In this article, we mainly discuss “pathways” as part of a cognitive model, i.e., as connections between representations. Still, it is likely that actual neuronal connections underlie these pathways.)

Our present study aims to dissociate the hypothesized graphemic and semantic pathways by comparing the time course of synesthetic activation through them. Different from previous studies which distinguished between perceptual and conceptual synesthesia, we are making this graphemic/semantic dissociation *within* conceptual synesthesia. Empirically, number–color synesthesia is consensually considered as conceptual/higher synesthesia, and the dissociation of graphemic and semantic pathways is a distinction of number–color synesthesia mechanisms. Theoretically, grapheme identities are considered as concepts in theorizations of synesthesia [14], and semantics is clearly conceptual rather than perceptual. Therefore, from both empirical and theoretical points of view, the graphemic/semantic dissociation is a dissociation within conceptual synesthesia. To the best of our knowledge, there have not been empirical studies on this dissociation. We hereby used numbers in different notations to dissociate the graphemic and semantic aspects of the synesthetic inducer. Specifically, numbers in graphemic notations (e.g., Arabic digits) activate synesthetic colors via both graphemic and semantic pathways, while numbers in non-graphemic notations (e.g., dots) activate colors via the semantic pathway alone. By varying the interval between onsets of the prime and target (i.e., the stimulus onset asynchrony, SOA) in a priming paradigm, we are able to measure synesthetic color activation (quantified by congruency effect) at different time points following the presentation of a number prime.

In our experiment, we used numbers in different notations (digits, dice, and non-canonical (NC) dot patterns) as the prime and manipulated the SOAs between the prime and target. Synesthetes and non-synesthetes reacted to a color patch target as quickly as possible. The SOA at which the congruency effect occurs, which we hereafter refer to as the latency of congruency effect, reflects the speed of synesthetic color activation. The maximum congruency effect size across SOA, which we hereafter refer to as the magnitude of congruency effect, reflects the strength of number–color association. Since graphemic processing precedes semantic processing [19,20,21], we expected that digits would elicit synesthetic color faster than dot patterns. In accordance with most synesthetes’ differential conscious experiences (digits are consciously experienced as colored, while dot patterns are not), we also expected that digits would activate synesthetic colors more strongly. Taken together, congruency effect yielded by digits should have a shorter latency and a larger magnitude than either dot notations. Additionally, participants may recognize (i.e., access the semantic numerosity of) dice patterns more rapidly than NC patterns due to more familiarity with dice, but both notations should lead to color activation via the semantic pathway. Therefore, we expect dice patterns to yield congruency effects with a shorter latency and similar magnitudes when compared with those in NC patterns.

## 2. Materials and Methods

### 2.1. Participants

Eleven synesthetes (4 males and 7 females, age range 18–26 years, *M* = 20.91, *SD* = 2.39) and eleven gender-matched non-synesthetes (age range 18–26 years, *M* = 20.91, *SD* = 2.66) participated in the experiment. Synesthete participants were then assessed and screened with a customized eight-item questionnaire. With this questionnaire, they reported several key features of synesthetic experience (see Appendix A for details). They were considered genuine synesthetes by meeting the following criteria: (1) they had a general feeling that numbers are colored; (2) their experiences of concurrent colors were involuntary; (3) their number–color pairings were consistent across time. They also gave verbal descriptions of their number–color pairings, which matched with their color choice in the experiment (see Section 2.4). This demonstration of consistency supported the genuineness of synesthesia. Non-synesthete participants denied having similar experiences as those of the synesthetes. All participants were right-handed. 

### 2.2. Stimuli and Apparatus

We used two kinds of stimuli in the experiment, the numbers and the color patches. A number stimulus was always colored black and occupied a 2° × 2° area at the center of the screen. We used three notations (Arabic digit, dice pattern, and NC dot pattern) for five numbers (2–6), the stimulus for a given number × notation was fixed, e.g., NC dot configuration for “4” did not change between trials (Figure 1). A color stimulus was a 2° × 2° square color patch appearing at the center of the screen. Specific colors differ between participants (see Section 2.4 for color selection procedure, and Appendix B for list of colors used).

The experiment was programmed with the Psychophysics Toolbox-3 [22,23,24] implemented on MATLAB 2018b (MathWorks Inc., Natick, MA, USA), on a 27-inch LCD computer screen. The refresh rate was 60 Hz. Participants rested their head on a chin rest, which kept their eyes aligned to the screen center at an 80 cm distance.

### 2.3. Design

We adopted a 3 (number notation) × 5 (SOA) × 2 (congruency) factorial design, yielding 30 experimental conditions. Number notations include Arabic digit, dice pattern, and NC pattern. SOAs were picked by a pilot test and set with five levels: 0, 100, 300, 400, and 800 ms. A trial was defined as “congruent” if the target color matched the corresponding color of the number prime and “incongruent” if they did not match.

We arranged the trials in 8 blocks. Only two numbers and their corresponding colors appeared in each block. Thus, a color (e.g., red or green) always had an equal probability (50%) of being either congruent or incongruent to its preceding number prime. Considering that numerical range (small or large numbers, usually taking 4 as the cut point) would affect the speed and accuracy of object numerosity processing [25] and may influence behavioral effects of synesthesia [7], we used numbers 2, 3, 5, and 6. We arranged them in a way to balance the numerical ranges (Table 1; except for the cases of four synesthetes and four non-synesthetes, in which number 4 was substituted for 3 or 5 in their experiments. See Section 3). Specifically, we included an equal number of trials with small and larger numbers, as well as an equal number of blocks where two numbers were in the same or different numerical ranges. Since some of our synesthetes reported associating numbers or colors to specific spatial layouts (e.g., feeling that certain colors belong to the left or right side), we therefore reversed the correspondence of left- and right-hand responses to colors in repetitions of identical blocks.

### 2.4. Procedure

Participants completed the experiment in two days, with four blocks (taking approximately 90 min) each day. Upon arrival at the laboratory on Day 1, they chose colors for numbers 2–6 from the custom font color palette in Microsoft Word (Microsoft Inc.). Synesthetes chose their synesthetic colors, and non-synesthetes assigned any color they preferred to the given numbers. In this way, though color sets were not physically identical across participant groups, they were functionally identical in that they were chosen by participants themselves.

Each block contained a color response practice session and a main experiment session. The practice session familiarized the participant with the correspondence of response keys (left and right “shift” on the keyboard) to colors. Participants were instructed to press the left “shift” key when seeing one color, and the right “shift” key when seeing the other. This response should be as quick as possible. Experimental stimuli were presented against a white background. Each trail started with a 300 ms central fixation cross (diameter: 0.6°, RGB: (128, 128, 128)), and a color stimulus was presented at screen center until response (maximum waiting time: 2000 ms). Immediately after response, a 500 ms “correct/incorrect” feedback was given. The next trial started after an inter-trial interval (ITI) of 1000 ms. There were 80 randomized trials (40 for each color).

The main experiment followed the practice session. Participants were instructed to respond to colors with the same keys as in the practice session, and then report numbers by pressing “Y” and “B” keys (the keys were labeled with numbers; “Y/B” always corresponded to the smaller/larger number in the block). The response to color (hereafter referred to as the color response) should be as quick and accurate as possible, and the response to number (hereafter referred to as the number response) could be at their own pace. In each trial, a central fixation cross (diameter: 0.6°, RGB: (128, 128, 128)) was presented for 300 ms, and a number stimulus was presented at screen center for 50 ms. After a delay (for SOA > 0 conditions), a color stimulus was presented at screen center until color response (for SOA = 0 ms, the number stimulus was superimposed on the color patch; maximum waiting time: 2000 ms). Immediately following color response, a prompt for number response was presented until the response was made (maximum waiting time: 5000 ms). No feedback was given throughout the trial. The next trial started after a random ITI of 2000–2500 ms (Figure 2).

Within a block, identical trials (i.e., trials with same condition and number) were repeated 3 times. Thus, each block contained 30 (conditions) × 2 (numbers) × 3 (repetition) = 180 trials. These trials were randomized. In total, the experiment contained 8 × 180 = 1440 trials.

## 3. Results

The color patches for each individual are listed in Appendix B (Table A1). There were four synesthetes who either had similar colors associated with two numbers or had black or white concurrent colors for 3 or 5. To ensure that the corresponding colors were easily distinguishable and could be seen against a white background and that a black number stimulus could be seen when superimposed on the color patch, 3 or 5 was substituted for 4 in their experiments. Correspondingly, the same number changes were made in four non-synesthetes’ experiments.

Four blocks were excluded due to technical failure. Two more were excluded due to number response accuracy lower than 3 *SD*s below average, which was thought to indicate inattentiveness. Trials with incorrect responses for either color or number were excluded from analysis. Within each participant, we removed outlier trials (where RT of color response was more than 3 *SD*s deviant from the mean RT). In all, 7.66% of trials in the remaining blocks were excluded. To check for the congruency effect, using SPSS 25.0 (IBM Corporation), we ran a repeated measures ANOVA on RT with participant group (i.e., synesthete or non-synesthete) as the between-subject factor, and congruency, number notation, and SOA as within-subject factors. Overall RTs of synesthetes (*M* ± *SE*: 550 ± 28 ms) and non-synesthetes (593 ± 28 ms) were not significantly different (*MD* = 43 ms, *F*(1,20) = 1.212, *p* = 0.284). The RTs for congruent trials (551 ± 18 ms) were significantly shorter than incongruent trials (592 ± 22 ms; *MD* = 41 ms, *F*(1,20) = 10.868, *p* = 0.004, partial η^2^ = 0.352), i.e., there was a congruency effect. There was an interaction between participant group and congruency (*F*(1,20) = 6.590, *p* = 0.018, partial η^2^ = 0.248). Congruency effects were significant in synesthetes (Bonferroni-corrected post-hoc comparison: *MD* = 73 ms, *p* < 0.001) but not in non-synesthetes (*MD* = 9 ms, *p* = 0.612). These results indicate a synesthetic congruency effect only in synesthetes, typical in synesthetic priming paradigms (Figure 3).

For each participant and under each condition, we calculated congruency effects by subtracting the average RT of congruent trials from the average RT of incongruent trials. We ran a repeated measures ANOVA on congruency effect with participant group as the between-subject factor, and number notation and SOA as within-subject factors. We found a significant main effect of participant group (synesthete (73 ± 18 ms) > non-synesthete (9 ± 18 ms), *MD* = 64 ms, *F*(1,20) = 6.590, *p* = 0.018, partial η^2^ = 0.248). SOA had a main effect (*F*(4,80) = 4.944, *p* = 0.001, partial η^2^ = 0.198) due to a larger congruency effect at 300 ms (60 ± 16 ms) than 800 ms (20 ± 8 ms) SOA (Bonferroni-corrected post-hoc comparison: *MD* = 40 ms, *p* = 0.016). The interaction between SOA and participant group was also significant (*F*(4,80) = 2.500, *p* = 0.049, partial η^2^ = 0.111). Bonferroni-corrected post-hoc comparison indicated that synesthetes’ congruency effects were larger at SOA of 100 ms (83 ± 21 ms; *MD* = 46 ms, *p* = 0.028) and 300 ms (104 ± 23 ms; *MD* = 67 ms, *p* = 0.003) than at 800 ms (37 ± 11 ms), whereas non-synesthetes’ congruency effects did not differ across SOAs. Number notation exhibited no main effect (*F*(2,40) = 2.095, *p* = 0.136) but interacted with SOA (*F*(8,160) = 2.178, *p* = 0.032, partial η^2^ = 0.098). At 400 ms SOA, congruency effect for digits (60 ± 17 ms) were larger than for dice (33 ± 13 ms; *MD* = 26 ms, *p* = 0.032), and at 800 ms SOA, congruency effect for NC patterns (38 ± 9 ms) were larger than for dice (5 ± 10 ms; *MD* = 0.33 ms, *p* = 0.007).

To determine the origin of interactions, we ran repeated measures ANOVAs with number notation and SOA as within-subject factors, separately in synesthetes and non-synesthetes. In synesthetes (Figure 4a), SOA had a main effect (*F*(4,40) = 5.450, *p* = 0.001, partial η^2^ = 0.353) due to larger congruency effect at 300 ms (73 ± 29 ms) than 800 ms (37 ± 13 ms) (Bonferroni-corrected post-hoc comparison: *MD* = 67 ms, *p* = 0.038). This is in line with our expectation that synesthetic interference would occur at short SOAs and later decline. Number notation had a marginally significant main effect (Greenhouse–Geissler-corrected for violation of sphericity assumption, *F*(1.287,12.873) = 4.332, *p* = 0.050, partial η^2^ = 0.302) due to larger congruency effect for digits (87 ± 27 ms) than for dice (64 ± 25 ms; Bonferroni-corrected post-hoc comparison: *MD* = 23 ms, *p* = 0.005). SOA × number notation interaction was marginally significant (*F*(8,80) = 1.960, *p* = 0.062, partial η^2^ = 0.164). Bonferroni-corrected simple main effect analysis revealed that at 0 ms SOA, congruency effect for digits (109 ± 26 ms) was larger than for NC patterns (44 ± 25 ms; *MD* = 0.65 ms, *p* = 0.018); at 400 ms SOA, congruency effect for digits (93 ± 32 ms) was larger than for dice (54 ± 24 ms; *MD* = 39 ms, *p* = 0.045); and at 800 ms SOA, congruency effect for NC patterns (56 ± 11 ms) was larger than for dice (17 ± 17 ms; *MD* = 0.39 ms, *p* = 0.033). Taken together, digits tend to yield larger congruency effects than dot patterns at 0 and 400 ms SOAs, while at SOA levels in between (100 and 300 ms), effect sizes were similar across number notations. The congruency effect elicited by digits was persistent throughout the initial 400 ms following stimulus onset, while the congruency effect elicited by dot patterns increased through 0–300 ms SOAs, but decreased from 300 ms onwards, i.e., exhibited a pattern of “rise-and-fall”. In non-synesthetes (Figure 4b, none of the effects were significant (number notation: *F*(2,20) = 0.124, *p* = 0.884; SOA: *F*(4,40) = 1.280, *p* = 0.294; interaction: *F*(8,80) = 0.823, *p* = 0.0585).

## 4. Discussion

In order to dissociate graphemic and semantic components in number–color synesthesia, we investigated the time course of color activations elicited by numbers in three notations. We implemented a priming paradigm with varying SOAs and compared number–color congruency effects. Throughout the 0–400 ms time window post-stimulus onset when differences between notations were expected to appear, we found differences between digits and dot patterns but not between types of dot patterns (i.e., dice and NC). Specifically, congruency effect elicited by digits had a shorter latency (i.e., an immediate onset) and was persistent, while that elicited by both dot patterns exhibited a “rise-and-fall” pattern. At 800 ms SOA, NC yielded an unexpected larger effect than dice. The three number notations yielded similar congruency effect sizes (i.e., maximum congruency effect sizes across SOA). Our findings revealed a distinction between effects of graphemic and semantic pathways of synesthesia on the temporal dimension. In contrast to previous studies distinguishing “perceptual vs conceptual” synesthesia, we highlighted that this distinction is within “conceptual” synesthesia. Moreover, the associated two pathways (as well as multiple working mechanisms) could co-exist in individual synesthetes, rather than a between-individual difference or a single mechanism, which could explain the current results (see more details below).

As we have expected, digits did show superiority over dot patterns in eliciting synesthesia, but this superiority was reflected in a quicker and more persistent time course rather than a larger magnitude. Thus, the graphemic and semantic pathways differ in time courses in their effects: effect of the former was quick and persistent, while the latter had actions of a relatively slow and transient manner. Previous studies have measured number–color synesthetic interference using either simultaneous stimuli presentation or a fixed SOA [2,6,7,8,9], providing information only at a single time point. Therefore, congruency effect sizes measured in these studies were interfered with by different latencies across conditions and may not accurately reflect “strengths” of synesthetic connections. Our study teased apart magnitudes and latencies of congruency effects and provided a dynamic and more comprehensive profile of synesthetic activation. The result of comparison between dice and NC patterns differed from our expectation. One possible explanation is that synesthetic color activation elicited by NC patterns had a longer latency, due to slower number recognition [Indeed, we tested the same group of participants for number recognition speed. Participants were slower in recognizing numbers from NC patterns than from dice patterns, and both were slower than that from digits (see Appendix C for details). Note that the difference between digit and dot patterns cannot be explained by different recognition speed alone. Faster recognition speed may account for the quick onset of congruency effect for digits, but not for the persistence. The observed difference in patterns (rather than mere latencies) of congruency effects across SOAs suggests a mechanism difference]. At 800 ms SOA, the congruency effect elicited by NC patterns had not completely decayed, while that by dice patterns had, resulting in a larger congruency effect of NC patterns at 800 ms SOA. It is noteworthy that congruency effects of digit (which activates colors via both pathways) and dot patterns (which activate colors via the semantic pathway alone) at 300 ms SOA were equivalent, suggesting that effects from the graphemic and semantic pathways are non-additive. A possible explanation of this non-additivity is that both pathways activate the same color representation (e.g., a common group of neurons, instead of different groups). When a digit is presented to a synesthete, this color representation is fully activated under the input from the graphemic pathway (e.g., causing maximum firing rate in the neurons) [26]. Additional input from the semantic pathway cannot further increase the activation level and, thus, do not contribute more to the behavioral outcome, namely the congruency effect.

Our distinction of pathways differs from previous studies in two major aspects. First, models of synesthesia typically distinguish between perceptual and conceptual levels of stimulus processing, and either argue for the occurrence of synesthesia on one of the levels (e.g., the conceptual mediation model [14]), or identify types of synesthesia based on its level of occurrence (e.g., lower and higher synesthesia [11,27]). Here, we are not making a parallel distinction; rather, we propose the distinction between graphemic and semantic aspects as a further distinction within conceptual synesthesia. Second, studies typically discussed multiple mechanisms of synesthesia as individual differences, possibly underlying phenomenological differences in synesthetic experiences (e.g., the spatial location where a color concurrent appears [12,28,29]). Here, by distinguishing the graphemic and semantic pathways, we are not proposing a further division of conceptual synesthesia into more detailed subtypes. Rather, we propose that the pathways are different mechanisms underlying a single type of synesthesia (i.e., conceptual synesthesia) and are co-existent in individual synesthetes. For the line of research on neural substrates of synesthesia, this implies that multiple pathways may be involved. For instance, the cross-activation model suggests that synesthesia arises from direct, local anatomical links between different modality-specific brain areas, and these links are unique to synesthetes [10,27,30,31]. Meanwhile, the hyperbinding account proposes that synesthesia results from global changes in the functioning of common brain structures. Specifically, when synesthetes’ brains process inducers, feedback from higher-level areas was less inhibited, resulting in additional signals corresponding to concurrents [1]. We suggest these accounts may not be mutually exclusive. Instead, different pathways can co-exist in individual synesthete brains, and may be accounted for by different neural mechanisms. Indeed, neuroimaging studies have provided both structural [32] and functional [33] findings supporting this view of pathway co-existence.

In sum, our study did not aim to separate different subtypes of number–color synesthesia but to establish a distinction between the graphemic and semantic aspects of number–color synesthesia on the temporal dimension. We revealed the temporal dynamics of the behavioral outcomes in number–color synesthesia, in which the graphemic pathway was quick and relatively persistent in sub-serving the color activation, while the semantic pathway had a slower onset and is more transient. Moreover, effects from the pathways appeared non-additive, possibly implying a common neuronal representation of the synesthetic concurrent. In contrast to previous studies which discussed the heterogeneity of synesthesia by distinguishing between its perceptual vs. conceptual nature and considered each type of synesthesia as a unitary whole, our distinction between graphemic and semantic pathways is a distinction within conceptual synesthesia. Moreover, the two pathways are co-existing mechanisms within individuals, rather than between-individual differences. For future studies, our study has demonstrated that investigating the time courses of synesthetic activation can be an effective methodology of distinguishing between different mechanisms in synesthesia; theoretically, we call attention to the possibility that a single type of synesthesia may involve multiple mechanisms.

## Figures and Tables

**Figure 1 brainsci-12-01400-f001:**
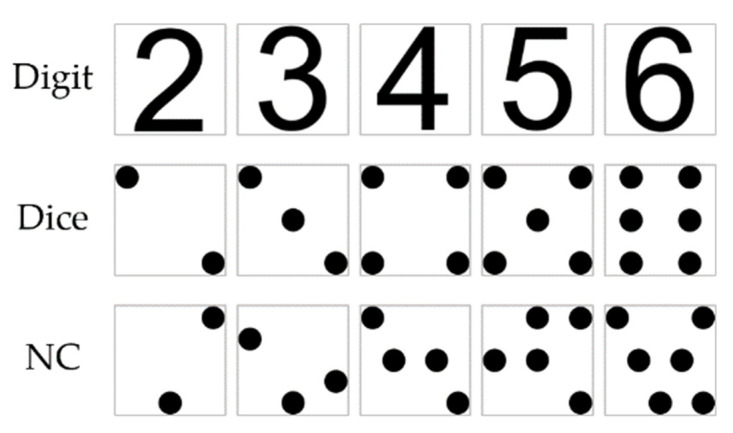
Three types of number stimuli for numbers 2–5. Grey lines delineate borders of stimuli area (2° × 2°, for illustration only), and were not present in the experiment.

**Figure 2 brainsci-12-01400-f002:**
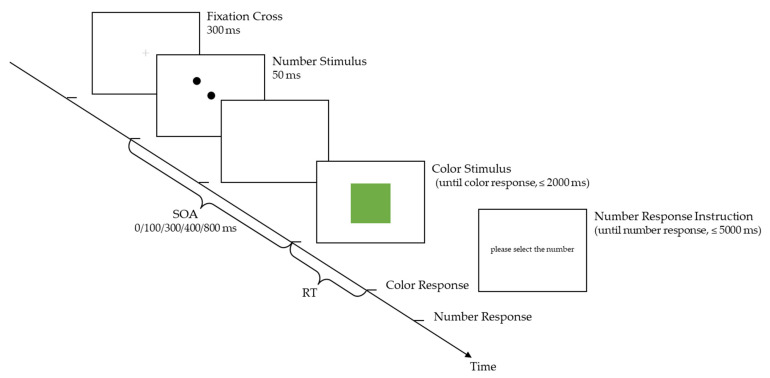
Schematic diagram of a main experiment trial.

**Figure 3 brainsci-12-01400-f003:**
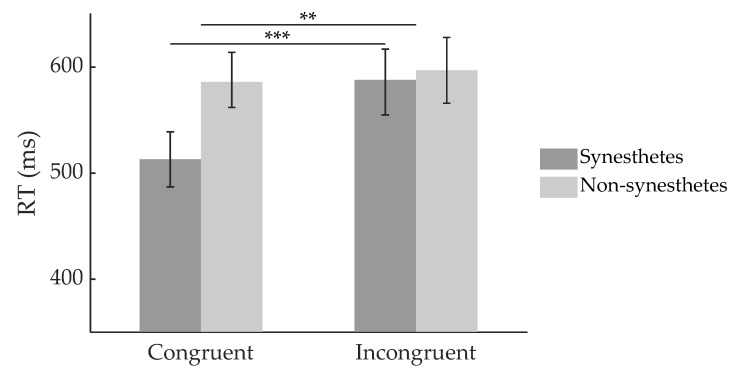
RTs of synesthetes and non-synesthetes in congruent and incongruent trails. Error bars denote one standard error. ** *p* < 0.01, *** *p* < 0.001.

**Figure 4 brainsci-12-01400-f004:**
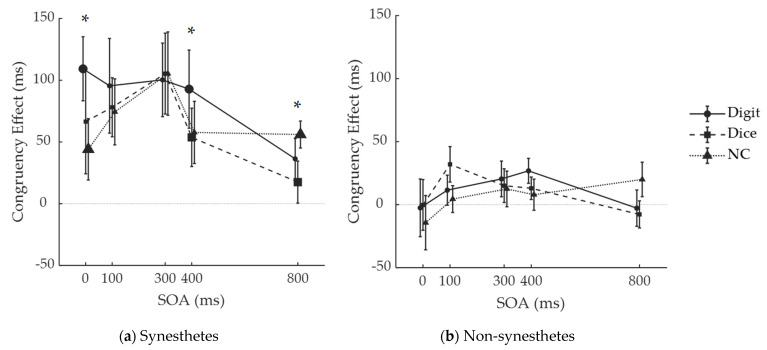
Congruency effect of three number notations as a function of SOA, for (**a**) synesthetes; (**b**) non-synesthetes. Error bars denote one standard error. The stars (*) indicate *p* < 0.05 for the test of simple main effects. The conditions where congruency effects were significantly different were marked with enlarged symbols.

**Table 1 brainsci-12-01400-t001:** Arrangement of numbers and hands for responses to colors in each block (a number in a color column denotes the color corresponding to that number). For participants who used other number sets (2, 3, 4, 6 or 2, 4, 5, 6), 4 took the place of 5 or 3.

	Block	Numbers	Colors to Which Participants Responded with Left- or Right-Hand Key-Press	
			Left Hand	Right Hand
Day 1	1	2, 3	2	3
	2	5, 6	6	5
	3	2, 5	5	2
	4	3, 6	3	6
Day 2	5	3, 6	6	3
	6	2, 5	2	5
	7	5, 6	5	6
	8	2, 3	3	2

## Data Availability

Data will be granted access to: https://osf.io/9z54d/ (accessed on 24 August 2022).

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
