# Peer review of "Graphemic and Semantic Pathways of Number–Color Synesthesia: A Dissociation of Conceptual Synesthesia Mechanisms"

_brainsci, 2022, doi:10.3390/brainsci12101400_

Round 1

Reviewer 1 Report

Summary

In this paper, the authors aim to tease apart direct, ‘digit-colour’ synaesthesia from more semantic numerosity related synaesthesia (‘semantic-synaesthesia’). They do this by using different types of colour inducing stimuli representing magnitude (digits, dice patterns, and dot patterns) that are followed by a coloured patch that is either congruent or incongruent with the inducer. By varying the SOA of the coloured patch onset, the authors aim to tease apart the time courses of digit and semantic synaesthesia. I feel the paper suffers from problems in the experimental design that make the data difficult to interpret.

Major comments

In the methods section, it is stated that the synaesthetes were tested for consistency, but this was not done using a standard consistency test with many items and 3 repetitions of each stimulus (e.g., Eagleman at all, BRM, 2007). Rather, the synaesthetes were asked to describe their colours verbally only for the numbers 2-6, and these descriptions were later checked with the chosen colours by the researchers. This is actually a very minimal and not very reliable test of synaesthetic consistency, effectively leaving the assessment of synaesthesia to be by self-report.

Control participants picked their own colours for the letter stimuli, and this means the colours used in the experiment were not the same for synaesthetes and controls. Usually, one control is yoked to one synaesthete to avoid group differences in the stimulus materials. Now, we cannot know whether any differential congruency effects were in place for controls, due to the different stimulus colours.  

As stated in the Appendix C and in the discussion section, processing times for numbers, dice patterns and dot patterns do differ. These different processing times make the interpretation of the SOA results very difficult – it is basically a confounding factor. How can we know any effects of SOA are really due to a longer colour activation time, if the differences could also be related to longer processing of the stimuli?  

In the data analysis, the RT congruency effects are used as the dependent variable. This means that first the difference in RT is calculated between the congruent and incongruent conditions, and only this difference score is than taken into a between-subjects analysis (that also includes number notation and SOA as factors). Although I understand that running a full ANOVA with group x congruency x number notation x SOA is a very large analysis that may be difficult to grasp, what is happening now is that actually, the raw RT values across groups are not even shown in the results section. It also means that we as readers cannot judge the relative size of the congruency effect: 72 ms is substantial if the RTs are ~600 ms, but if they are closer to ~1500 ms, perhaps the 73 ms difference is less relevant.

I would like to see a table or graph with the raw RTs for each condition, or at least for congruent vs incongruent trials per condition per group, and I would like to see a verification that the between-groups effect is still there if not the congruency effect, but the raw RTs are taken into a group x congruency x number notation x SOA ANOVA.

Minor comments

p. 1, line 35. “In particular, in grapheme-color synesthesia, inducers are graphemes (e.g., letters, digits) and concurrents are colors, while in number-color synesthesia, inducers are 36 numbers and concurrents are colors.”

I think these examples are rather confusing, because strictly speaking number-color synaesthesia is also already part of grapheme-color synaesthesia (since graphemes are all written symbols such as letters and numbers/digits).

Author Response

Q1: Test for synesthetic consistency was inadequate.

A1: The widely-used Synesthesia Battery (Eagleman et al., 2007) was currently unavailable on the website. Indeed, we did base our assessment of synesthesia largely on self-report, but we questioned participants on multiple and comprehensive phenomenological characteristics of synesthesia, including automaticity, consistency, clear distinction of synesthetic and physical colors, etc. With these, we aim to ensure that participants’ understanding of synesthesia was accurate, and that their self-report was truthful, meanwhile to largely avoid informing them the “right” phenomenology of synesthesia.

Q2: Synaesthetes and non-synaesthetes were not yoked. Non-synaesthetes chose their own, different color sets for the numbers.

A2: In previous studies, a non-synaesthete was typically yoked to a synaesthete, and used same inducer-concurrent pairings that the synaesthete chose. This also means that for synaesthetes, colors were self-assigned, while for non-synaesthetes, colors were assigned by the experimenter. Whether the experimental materials were assigned by the participant him/herself may affect performance, as shown the self-generation effect in memory (self-generated materials were memorized better than assigned ones, Slamecka & Graf, 1978). Specifically, the practice in typical synesthesia research may result in synaesthetes being more aware of the inducer-concurrent correspondence (because of self-assignment instead of synesthesia), and thus facilitate congruency effects of synaesthetes. Controlling for this factor inevitably resulted in using different colors between groups. However, color choices varied widely between individuals as seen in Appendix B. If specific colors did affect RTs or congruency effects differently, the overall effect could still be cancelled out in group-level averaging (though admittingly, this was not a systematic control).

In sum, there were two aspects of the color stimuli: (1) what the specific color was, and (2) whether it was assigned by the participant him/herself or the experimenter. It was not possible to achieve precise control of both factors. We reasoned that the latter was more problematic (and potential influential factor) for our experiment, so we controlled for it by letting non-synaesthetes choose their own colors.

Q3: Different stimulus processing speed may confound with synesthetic color activation speed across number notations.

A3: Congruency effects elicited by digits appeared quicker (digit > NC at 0 ms SOA) and was more persistent (digit > dice at 400 ms SOA) than that elicited by dot patterns. While the “quicker” aspect could be explained by faster recognition of digits, the “more persistent” aspect could not. This pattern difference led us to propose a difference in color activation mechanisms between digit and dot notations. If the effect of SOA were only due to different number recognition speed, the difference in time courses of congruency effects across number notations should demonstrate simple horizontal shifts (i.e., differences in latency), which was not the case. 

Q4: Raw RTs should be presented and analyzed.

A4: We have added relevant information in the revised text (current Figure 3).

Q5: Number-color synaesthesia is also already part of grapheme-color synaesthesia

A5: Conceptually speaking, we accept this suggestion. However, the current findings suggested that different notations of numbers make difference, a strict empirical test and comparison between number and grapheme should be implemented in future studies.

Eagleman, D. M., Kagan, A. D., Nelson, S. S., Sagaram, D., & Sarma, A. K. (2007;2006;). A standardized test battery for the study of synesthesia. Journal of Neuroscience Methods, 159(1), 139-145. https://doi.org/10.1016/j.jneumeth.2006.07.012

Slamecka, N. J., & Graf, P. (1978). The generation effect: Delineation of a phenomenon. Journal of Experimental Psychology. Human Learning and Memory, 4(6), 592-604. https://doi.org/10.1037/0278-7393.4.6.592

Reviewer 2 Report

This is a very interesting study that is well written with a sound methodology. It helps to refine models of number-color synesthesia, as a re-evaluation of the emphasis on synesthesia as a perceptual phenomenon is needed.

However, I do have a concern over the (too) strong statement made by the authors on the conceptual nature of number-color synesthesia. While there is no doubt that graphemic and semantic pathways are parts of number-color synesthesia mechanisms, I am not totally convinced that their nature are both conceptual. The arguments raised by authors against perceptual (vs. conceptual) are weak (while the conceptual nature of the semantic part is obvious). It is not because that inducers and concurrents are linked on the conceptual level that the classical distinction between perceptual and semantic /conceptual synesthesia doesn’t exist. While I agree with the authors that distinguishing graphemic vs. semantic pathways might be more helpful (and less confusing), their statement that the graphemic/semantic dissociation is within conceptual synesthesia is not well supported (either by previous studies or the data of the paper). I am not against the authors conception, but I don’t think that the current knowledge allows us to get rid of perceptual synesthesia (and the sub-types of synesthesia).  I would appreciate if the authors could be a little more nuanced in that matter throughout the paper (e.g., revising the title of the paper according – by removing the word conceptual for instance – and the interpretations of the results).  

Minor point:

P6/Results: It is not clear to me how  “7.66% of all trials were excluded” while it is written that “four blocks of trials… were excluded “ (each block seems to contain 180 trials according to the last paragraph of the Method section). Please clarify/revise.

Author Response

Q1: The claim that number-color synesthesia is conceptual in nature is too strong; perceptual synesthesia couldn’t be abandoned.

A1: We did not intend to abandon “perceptual synesthesia”, but to confine our research to the “conceptual” realm. We have revised the language to clarify this and avoid misunderstanding (Lines 67,68,71,103)

Q2: P6/Results: It is not clear to me how  “7.66% of all trials were excluded” while it is written that “four blocks of trials… were excluded “ (each block seems to contain 180 trials according to the last paragraph of the Method section). Please clarify/revise.

A2: Thanks for careful examination. 7.66% refers to the outliers of remaining trials (lines 239-242).

Reviewer 3 Report

Review of Graphemic and Semantic Pathways of Number-Color Synesthesia: A Dissociation of Conceptual Synesthesia Mechanisms

This is a well-designed study that uses a nice SOA manipulation to address the time course of different forms of synesthesia. The authors found that Arabic numbers produces rapid congruency effects, while other forms of numerical representations were somewhat delayed but eventually equally robust. This finding appears novel in the literature.

I found the analyses done competently and the data generally supportive of the conclusions. I guess my one complaint is that the number of subjects is on the low end, but I imagine these subjects are hard to recruit.

In looking at Table S2, are there consistencies noted by the authors? For example, the color choices for 1 and 5 seem darker than the rest.

Other than this, I don’t have any major issues with the study or the conclusions.

I sign all reviews (where allowed by the journal),

Author Response

Q: There seemed to be signs of inter-individual consistency of color choices.

A: Thanks for careful observation, we may instinctively notice such signs from Table S2. These trends are interesting, but meaningful analysis will require big data which is beyond our scope (e.g., Rich et al., 2005). Overall, our results did not support a bias of RTs for certain colors and the effects (if any) could be cancelled out with current randomization procedure.

Rich, A. N., Bradshaw, J. L., & Mattingley, J. B. (2005). A systematic, large-scale study of synaesthesia: Implications for the role of early experience in lexical-colour associations. Cognition, 98(1), 53-84. https://doi.org/10.1016/j.cognition.2004.11.003